# New Bounds For Distributed Mean Estimation and Variance Reduction

**Peter Davies**
IST Austria
peter.davies@ist.ac.at

**Vijaykrishna Gurunathan**
IIT Bombay
krishnavijay1999@gmail.com

**Niusha Moshrefi**
IST Austria
niusha.moshrefi@ist.ac.at

**Saleh Ashkboos**
IST Austria
saleh.ashkboos@ist.ac.at

**Dan Alistarh**
IST Austria & NeuralMagic
dan.alistarh@ist.ac.at

## Abstract

We consider the problem of *distributed mean estimation (DME)*, in which $n$ machines are each given a local $d$-dimensional vector $\boldsymbol{x}_v \in \mathbb{R}^d$, and must cooperate to estimate the mean of their inputs $\boldsymbol{\mu} = \frac{1}{n} \sum_{v=1}^n \boldsymbol{x}_v$, while minimizing total communication cost. DME is a fundamental construct in distributed machine learning, and there has been considerable work on variants of this problem, especially in the context of *distributed variance reduction* for stochastic gradients in parallel SGD. Previous work typically assumes an upper bound on the norm of the input vectors, and achieves an error bound in terms of this norm. However, in many real applications, the input vectors are concentrated around the correct output $\boldsymbol{\mu}$, but $\boldsymbol{\mu}$ itself has large norm. In such cases, previous output error bounds perform poorly.

In this paper, we show that output error bounds need not depend on input norm. We provide a method of quantization which allows distributed mean estimation to be performed with solution quality dependent only on the *distance between inputs*, not on input norm, and show an analogous result for distributed variance reduction. The technique is based on a new connection with lattice theory. We also provide lower bounds showing that the *communication to error trade-off* of our algorithms is asymptotically optimal. As the lattices achieving optimal bounds under $\ell_2$-norm can be computationally impractical, we also present an extension which leverages easy-to-use cubic lattices, and is loose only up to a logarithmic factor in $d$. We show experimentally that our method yields practical improvements for common applications, relative to prior approaches.

## 1 Introduction

Several problems in distributed machine learning and optimization can be reduced to variants *distributed mean estimation* problem, in which $n$ machines must cooperate to jointly estimate the mean of their $d$-dimensional inputs $\boldsymbol{\mu} = \frac{1}{n} \sum_{v=1}^n \boldsymbol{x}_v$ as closely as possible, while minimizing communication. In particular, this construct is often used for *distributed variance reduction*: here, each machine receives as input an independent probabilistic estimate of a $d$-dimensional vector $\boldsymbol{\nabla}$, and the aim is for all machines to output a *common estimate* of $\boldsymbol{\nabla}$ with *lower variance* than the individual inputs, minimizing communication. Without any communication restrictions, the ideal output would be the mean of all machines' inputs.

While variants of these fundamental problems have been considered since seminal work by Tsitsiklis & Luo (1987), the task has seen renewed attention recently in the context

of distributed machine learning. In particular, variance reduction is a key component in data-parallel distributed stochastic gradient descent (SGD), the standard way to parallelize the training of deep neural networks, e.g. Bottou (2010); Abadi et al. (2016), where it is used to estimate the average of gradient updates obtained in parallel at the nodes. Thus, several prior works proposed efficient compression schemes to solve variance reduction or mean estimation, see e.g. Suresh et al. (2017); Alistarh et al. (2017); Ramezani-Kebrya et al. (2019); Gandikota et al. (2019), and Ben-Nun & Hoefler (2019) for a general survey of practical distribution schemes. These schemes seek to *quantize* nodes' inputs coordinate-wise to one of a limited collection of values, in order to then efficiently encode and transmit these quantized values. A trade-off then arises between the *number of bits sent*, and the *added variance* due of quantization.

Since the measure of *output* quality is variance, it appears most natural to evaluate this with respect to *input* variance, in order to show that *variance reduction* is indeed achieved. Surprisingly, however, we are aware of no previous works which do so; all existing methods give bounds on output variance in terms of the *squared input norm*. This is clearly suboptimal when the squared norm is higher than the variance, i.e., when inputs are not centered around the origin. In some practical scenarios this causes output variance to be *higher* than input variance, as we demonstrate in Section 4.

**Contributions.** In this paper, we provide the first bounds for distributed mean estimation and variance reduction which are still tight when inputs are not centered around the origin. Our results are based on new lattice-based quantization techniques, which may be of independent interest, and come with matching lower bounds, and practical extensions. More precisely, our contributions are as follows:

- For *distributed mean estimation*, we show that, to achieve a reduction of a factor $q$ in the input 'variance' (which we define to be the maximum squared distance between inputs), it is necessary and sufficient for machines to communicate $\Theta(d \log q)$ bits.

- For *variance reduction*, we show tight $\Theta(d \log n)$ bounds on the worst-case communication bits required to achieve optimal $\Theta(n)$-factor variance reduction by $n$ nodes over $d$-dimensional input, and indeed to achieve any variance reduction at all. We then show how incorporating *error detection* into our quantization scheme, we can also obtain tight bounds on the bits required *in expectation*.

- We show how to efficiently instantiate our lattice-based quantization framework in practice, with guarantees. In particular, we devise a variant of the scheme which ensures close-to-optimal communication-variance bounds even for the standard *cubic lattice*, and use it to obtain improvements relative to the best known previous methods for distributed mean estimation, both on synthetic and real-world tasks.

## 1.1 Problem Definitions and Discussion

MeanEstimation is defined as follows: we have $n$ machines $v$, and each receives as input a vector $\boldsymbol{x}_v \in \mathbb{R}^d$. We also assume that all machines receive a common value $y$, with the guarantee that for any machines $u, v$, $\|\boldsymbol{x}_u - \boldsymbol{x}_v\| \leq y$. Our goal is for all machines to output the same value $\boldsymbol{EST} \in \mathbb{R}^d$, which is an unbiased estimator of the mean $\boldsymbol{\mu} = \frac{1}{n} \sum_{v \in M} \boldsymbol{x}_v$, i.e. $\mathbf{E}[\boldsymbol{EST}] = \boldsymbol{\mu}$, with variance as low as possible. Notice that the input specification is entirely deterministic; any randomness in the output arises only from the algorithm used.

In the variant of VarianceReduction, we again have a set of $n$ machines, and now an unknown *true* vector $\boldsymbol{\nabla}$. Each machine $v$ receives as input an independent unbiased estimator $\boldsymbol{x}_v$ of $\boldsymbol{\nabla}$ (i.e., $\mathbf{E}[\boldsymbol{x}_v] = \boldsymbol{\nabla}$) with variance $\mathbf{E}[\|\boldsymbol{x}_v - \boldsymbol{\nabla}\|^2] \leq \sigma^2$. Machines are assumed to have knowledge of $\sigma$. Our goal is for all machines to output the same value $\boldsymbol{EST} \in \mathbb{R}^d$, which is an unbiased estimator of $\boldsymbol{\nabla}$, i.e., $\mathbf{E}[\boldsymbol{EST}] = \boldsymbol{\nabla}$, with low variance. Since the input is random, output randomness now stems from this input randomness as well as any randomness in the algorithm.

VarianceReduction is common for instance in the context of gradient-based optimization of machine learning models, where we assume that each machine $v$ processes local samples in order to obtain a *stochastic gradient* $\tilde{g}_v$, which is an unbiased estimator of the true gradient $\boldsymbol{\nabla}$, with variance bound $\sigma^2$. If we directly averaged the local stochastic gradients $\tilde{g}_v$, we

could obtain an unbiased estimator of the true gradient $G$ with variance bound $\sigma^2/n$, which can lead to faster convergence.

**Input Variance Assumption.** The parameter $y$ replaces the usual MEANESTIMATION assumption of a known bound $\mathbb{M}$ on the *norms of input vectors*. Note that, in the worst case, we can always set $y = 2\mathbb{M}$ and obtain the same asymptotic upper bounds as in e.g. Suresh et al. (2017); our results are therefore *at least* as good as previous approaches in all cases, but provide significant improvement when inputs are not centered around the origin.

The reason for this change is to allow stronger bounds in scenarios where we expect inputs to be closer to each other than to the origin. In particular, it allows our MEANESTIMATION problem to more effectively generalize VARIANCEREDUCTION. Parameter $y$ is a deterministic analogue of the parameter $\sigma$ for VARIANCEREDUCTION; both $y$ and $\sigma$ provide a bound on the distance of inputs from their *mean*, rather than from the origin. Accordingly, *input variance $\sigma^2$* for a VARIANCEREDUCTION instance corresponds (up to constant factors) to $y^2$ for a MEANESTIMATION instance. For consistency of terminology, we therefore refer to $y^2$ as the *input variance* of the instance (despite such inputs being deterministic).

It is common in machine learning applications of VARIANCEREDUCTION to assume that an estimate of the variance $\sigma^2$ is known (Alistarh et al., 2017; Gandikota et al., 2019). To study both problems in a common framework, we make the analogous assumption about MEANESTIMATION, and assume knowledge of the *input variance $y^2$*. Even if the relevant bounds $y$ or $\sigma$ are not known *a priori*, they can usually be estimated in practice.

**Relationship Between Problems.** If one allows unrestricted communication, the natural solution to both problems is to average the inputs. This is an exact solution to MEANESTIMATION with variance 0, and is also an asymptotically optimal solution to VARIANCEREDUCTION, of variance at most $\frac{\sigma^2}{n}$.[1] However, doing so would require the exchange of infinite precision real numbers. So, it is common to instead communicate *quantized* values of bounded bit-length (Alistarh et al., 2017), which will engender additional variance caused by random choices within the quantization method. The resulting estimates will therefore have variance $\boldsymbol{Var}_{quant}$ for MEANESTIMATION, and $\frac{\sigma^2}{n} + \boldsymbol{Var}_{quant}$ for VARIANCEREDUCTION. We will show a trade-off between bits of communication and output variance for both problems; in the case of VARIANCEREDUCTION, though, there is an 'upper limit' to this trade-off, since we cannot go below $\Omega(\frac{\sigma^2}{n})$ total output variance.

The other major difference between the two problems is that in MEANESTIMATION, distances between inputs are bounded by $y$ with certainty, whereas in VARIANCEREDUCTION they are instead bounded by $O(\sigma)$ only in *expectation*. This causes extra complications for quantization, and introduces a gap between average and worst-case communication cost.

**Distributed Model.** We aim to provide a widely applicable method for distributed mean estimation, and therefore we avoid relying on the specifics of particular distributed models. Instead, we assume that the basic communication structures we use (stars and binary trees) can be constructed without significant overhead. This setting is supported by machine learning applications, which have very high input dimension (i.e., $d \gg n$), and so the costs of synchronization or construction of an overlay (which do not depend on $d$, and are generally poly-logarithmic in $n$), will be heavily dominated by the communication costs incurred subsequently during mean estimation. They also need only be incurred once, even if mean estimation or variance reduction is to be performed many times (e.g. during distributed SGD). For these reasons, we do not include these model-specific setup costs in our stated complexities; any implementation should take them into separate consideration.

For simplicity, we will present our algorithms within a basic synchronous fault-free message-passing model, in which machines can send arbitrary messages to any other machine, but they could naturally be extended to asynchronous and shared-memory models of communication. Our aim will be to minimize the number of bits sent and received by any machine during the algorithm; we do not consider other measures such as round complexity.

**Vector Norms.** When dealing with vectors in $\mathbb{R}^d$, we will use names in bold, e.g. $\boldsymbol{x}$, $\boldsymbol{y}$. We will state most of our results in such a way that they will apply to any of the three

---

[1]For specific classes of input distribution, and for non-asymptotic concentration results, however, better estimators of $\boldsymbol{\nabla}$ are known; see e.g. Joly et al. (2017).

most commonly-used norms on $\mathbb{R}^d$ in applications: $\ell_1$ norm $\|\boldsymbol{x}\|_1 := \sum_{i=1}^{d} |x_i|$, $\ell_2$ norm $\|\boldsymbol{x}\|_2 := \sqrt{\sum_{i=1}^{d} x_i^2}$, and $\ell_\infty$ norm $\|\boldsymbol{x}\|_\infty := \max_{i=1}^{d} \|x_i\|$. Throughout the paper we will therefore use the general notation $\|\cdot\|$, which should be considered to be fixed as one of these norms, other than for statements specific to particular norms. Definitions which depend on norms, such as variance $\mathbf{Var}\,[\boldsymbol{x}] := \mathbf{E}\,\left[\|\boldsymbol{x} - \mathbf{E}\,[\boldsymbol{x}]\|^2\right]$, are therefore assumed to also be under the appropriate norm.

## 1.2 Related Work

Several recent works consider efficient compression schemes for stochastic gradients, e.g. Seide et al. (2014); Wang et al. (2018); Alistarh et al. (2017; 2018); Stich et al. (2018); Wen et al. (2017); Wangni et al. (2018); Lu & Sa (2020). We emphasize that these works consider a related, but different problem: they usually rely on assumptions on the input structure—such as second-moment bounds on the gradients—and are evaluated primarily on the practical performance of SGD, rather than isolating the variance-reduction step. (In some cases, these schemes also rely on history/error-correction (Aji & Heafield, 2017; Dryden et al., 2016; Alistarh et al., 2018; Stich et al., 2018).) As a result, they do not provide theoretical bounds on the problems we consider. In this sense, our work is closer to Suresh et al. (2017); Konečný & Richtárik (2018); Gandikota et al. (2019), which focus primarily on the *distributed mean estimation* problem, with SGD as only one potential application.

For example, QSGD (Alistarh et al., 2017) considers a similar problem to VarianceReduction; the major difference is that coordinates of the input vectors are assumed to be specified by 32-bit floats, rather than arbitrary real values. Hence, transmitting input vectors exactly already requires only $O(d)$ bits. They therefore focus on reducing the constant factor (and thereby improving practical performance for SGD), rather than providing asymptotic results on communication cost. They show that the expected number of bits per entry can be reduced from 32 to 2.8, at the expense of having an output variance bound in terms of *input norm* rather than *input variance*.

This is a common issue with existing quantization schemes, which leads to non-trivial complications when applying quantization to gradient descent and variance-reduced SGD (Künstner, 2017) or to model-averaging SGD (Lu & Sa, 2020), since in this case the inputs are clearly not centered around the origin. The standard way to circumvent this issue, adopted by the latter two references, but also by other work on quantization (Mishchenko et al., 2019), is to carefully adapt the quantization scheme and the algorithm to remove this issue, for instance by quantizing *differences* with respect to the last quantization point. These approaches, however, do not provide improvement as 'one-shot' quantization methods, and instead rely on historical information and properties of SGD or the function to optimize (such as smoothness). They are therefore inherently application-specific. Our method, by contrast, does not require "manual" centering of the iterates, and does not require storage of previous iterates, or any properties thereof. Konečný & Richtárik (2018) study MeanEstimation under similar assumptions, and are the only prior work to use quantization centered around points *other* than the origin. However, again prior knowledge about the input distribution must be assumed for their scheme to provide any improvements.

Suresh et al. (2017) study the MeanEstimation problem defined on real-valued input vectors. They present a series of quantization methods, providing an $O(\frac{1}{n^2} \sum_{v \leq n} \|\boldsymbol{x}_v\|_2^2)$ upper bound, and corresponding lower bounds. Recent work by Gandikota et al. (2019) studies VarianceReduction, and uses multi-dimensional quantization techniques. However, their focus is on protocols using $o(d)$-bit messages per machine (which we show *cannot* reduce input variance). They do give two quantization methods using $\Theta(d)$-bit messages. Of these, one gives an $O(\frac{1}{n} \max_{v \leq n} \|\boldsymbol{x}_v\|_2^2)$ bound on output variance, similar to the bound of Suresh et al. (2017) for MeanEstimation (the other is much less efficient since it is designed to achieve a privacy guarantee). Mayekar & Tyagi (2020) obtain a similar error bound but with slightly longer $\Theta(d \log \log \log(\log^* d))$-bit messages.

All of the above works provide output error bounds based on the norms of input vectors. This is only optimal under the implicit assumption that inputs are centered around the origin. In Section 4 we provide evidence that this assumption does not hold in some practical

scenarios, where the input (gradient) variance can be much lower than the input (gradient) norm: intuitively, for SGD, input variance is only close to squared norm when true gradients are close to $\mathbf{0}$, i.e., the optimization process is already almost complete.

# 2 OUR RESULTS

In this work, we argue that it is both stronger and more natural to bound output variance in terms of input variance, rather than squared norm. We devise optimal quantization schemes for MEANESTIMATION and VARIANCEREDUCTION, and prove matching lower bounds, regardless of input norms. We summarize the main ideas that lead to these results.

## 2.1 LATTICE-BASED QUANTIZATION

The reason that all prior works obtain output variance bounds in terms of input norm rather than variance is that they employ sets of quantization points which are *centered* around the origin $\mathbf{0}$. We instead cover the entire space $\mathbb{R}^d$ with quantization points that are in some sense uniformly spaced, using *lattices*.

Lattices are subgroups of $\mathbb{R}^d$ consisting of the integer combinations of a set of basis vectors. It is well-known (Minkowski, 1911) that certain lattices have desirable properties for covering and packing Euclidean space, and lattices have been previously used for some other applications of quantization (see, e.g., Gibson & Sayood (1988)), though mostly only in low dimension. By choosing an appropriate family of lattices, we show that any vector in $\mathbb{R}^d$ can be rounded (in a randomized, unbiased fashion) to a nearby lattice point, but also that there are not too many nearby lattice points, so they can be specified using few bits.

Lattices contain an infinite number of points, and therefore any encoding using a finite number of bits must use bit-strings to refer to an infinite amount of lattice points. To allow the receiver in our quantization method to correctly decode the intended point, we utilize the fact that we have a bound on the distance between any two machines' inputs ($y$ for MEANESTIMATION, and $O(\sigma\sqrt{n})$ (probabilistically, by Chebyshev's inequality) for VARIANCEREDUCTION). Therefore, if all points that map to the same bit-string are sufficiently far apart, a machine can correctly decode based on proximity to its own input.

The simplest version of our lattice quantization algorithm can be described as follows

- To encode $\boldsymbol{x}_u$, randomly map to one of a set of nearby lattice points forming a convex hull around $\boldsymbol{x}_u$. Denote this point by $\boldsymbol{z}$.
- Send $\boldsymbol{z} \bmod q$ under the lattice basis: $q$ is the quantization precision parameter.
- To decode with respect to $\boldsymbol{x}_v$, output the closest lattice point to $\boldsymbol{x}_v$ matching $\boldsymbol{z} \bmod q$ .

By showing that $\boldsymbol{x}_u$ is contained within a convex hull of nearby lattice points, we can round to one of these points randomly to obtain $\boldsymbol{z}$ such that the expectation of $\boldsymbol{z}$ is $\boldsymbol{x}_u$ itself, thereby ensuring *unbiasedness*. This is because a point within a convex hull can be expressed as a linear combination of its vertices with coefficients in $[0, 1]$, which we can use as rounding probabilities.

Our reason for using $\bmod q$ with respect to the lattice basis in order to encode lattice points into bit-strings is that by exploiting properties of the particular lattices we employ, we can show a lower bound on the distance between points encoded with the same bit-string, while also controlling the number of bits we use. Then, since points encoded with the same string are sufficiently far apart, our proximity-based decoding procedure can determine the correct point. We also have a parameter $\epsilon$ which controls the granularity of the lattice used. This method of lattice-based quantization gives the following guarantee for communicating a vector between two parties:

**Theorem 1.** *For any $q = \Omega(1)$, any $\epsilon > 0$, and any two parties $u$, $v$ holding input vectors $\boldsymbol{x}_u$, $\boldsymbol{x}_v \in \mathbb{R}^d$ respectively, there is a quantization method in which $u$ sends $O(d \log q)$ bits to $v$, and if $\|\boldsymbol{x}_u - \boldsymbol{x}_v\| = O(q\epsilon)$, $v$ can recover an unbiased estimate $\boldsymbol{z}$ of $\boldsymbol{x}_u$ with $\|\boldsymbol{z} - \boldsymbol{x}_u\| = O(\epsilon)$.*

Details and proofs are deferred to the full version of this paper due to space constraints. This result is very general, and can be applied not only to distributed mean estimation but to any application in which high-dimensional vectors are communicated, in order to reduce communication.

## 2.2 Upper Bounds

Next, we show how to apply our quantization procedure to MeanEstimation and VarianceReduction. We wish to gather (quantized estimates of) machines' inputs to a single machine, which computes the average, and broadcasts a quantized estimate of this average to all other machines. For simplicity of analysis we do so using a star and a binary tree as our communications structures, but any connected communication topology would admit such an approach. The basic star topology algorithm can be described as follows:

- All machines $u$ send $\boldsymbol{x}_u$, quantized with precision parameter $q$ and $\epsilon = \Theta(y/q)$, to the leader machine $v$.
- Machine $v$ decodes all received vectors, averages them, and broadcasts the result, using the same quantization parameters.

By choosing the leader randomly we can obtain tight bounds in expectation on the number of communication bits used per machine, and by using a more balanced communication structure such as a binary tree we can extend these bounds in expectation to hold with certainty (at the expense of requiring more communication *rounds*).

**Theorem 2.** *For any $q = \Omega(1)$, MeanEstimation can be performed with each machine using strictly $O(d \log q)$ communication bits, with $O(\frac{y^2}{q})$ output variance.*

**Theorem 3.** *VarianceReduction can be performed using strictly $O(d \log n)$ bits, with $O(\frac{\sigma^2}{n})$ output variance, succeeding with high probability.*

These results give optimal communication-variance bounds for these problems. However, to make this bounds practical, we address two main challenges.

**Challenge 1: Input Variance.** One difficulty with our approach is that we assume a known estimate of input variance ($y^2$ for MeanEstimation, $\sigma^2$ for VarianceReduction). Furthermore, in VarianceReduction, even if our input variance estimate is correct, some pairs of inputs can be further apart, since the bound is probabilistic.

To address this problem, we develop a mechanism for *error detection*, which allows the receiver to detect if the encode and decode vectors ($\boldsymbol{x}_u$ and $\boldsymbol{x}_v$ respectively) are too far apart for successful decoding. In this way, if our estimate of input variance proved too low, we can increase either it or the number of bits used for quantization until we succeed. The main idea is that rather than using mod $q$ to encode lattice points, we use a more sophisticated *coloring* of the lattice and a new encoding procedure to ensure if $\boldsymbol{x}_u$ and $\boldsymbol{x}_v$ are far apart, with high probability the encoder $v$ chooses a color which is not used by any nearby point to $\boldsymbol{x}_u$, and therefore $u$ can tell that $\boldsymbol{x}_v$ was not nearby.

As an application, we obtain an algorithm for VarianceReduction which uses an optimal expected number of bits per machine (except for an additive $\log n$, which in our applications is assumed to be far smaller than $d \log q$):

**Theorem 4.** *For any $q = \Omega(1)$, VarianceReduction can be performed using $O(d \log q + \log n)$ communication bits per machine in expectation, with $O(\frac{\sigma^2}{q} + \frac{\sigma^2}{n})$ output variance, succeeding with high probability.*

**Challenge 2: Computational Tractability.** Another issue is that known lattices which are optimal for $\ell_1$ and $\ell_2$-norms can be computationally prohibitive to generate and use for problems in high dimension. We show that if we instead use the standard *cubic lattice*, which is optimal under $\ell_\infty$-norm and admits straightforward $\tilde{O}(d)$-computation encoding and decoding algorithms, in combination with a structured random rotation using the Walsh-Hadamard transform as proposed by Suresh et al. (2017), we can come within a log-factor variance of the optimal bounds of Theorems 2, 3, and 4.

**Theorem 5.** *Using the cubic lattice with a random rotation, we achieve the following output variances under $\ell_2$-norm (succeeding with high probability):*

- $O(\frac{y^2 \log nd}{q})$ *for* MEANESTIMATION, *using strictly* $O(d \log q)$ *bits;*

- $O(\frac{\sigma^2 \log d}{n})$ *for* VARIANCEREDUCTION, *using strictly* $O(d \log n)$ *bits;*

- $O(\frac{\sigma^2 \log nd}{q} + \frac{\sigma^2}{n})$ *for* VARIANCEREDUCTION, *using* $O(d \log q + \log n)$ *bits in expectation.*

*Furthermore, each machine need perform only $\tilde{O}(d)$ computation in expectation.*

Under the cubic lattice, our quantization method is particularly simple and efficient in practice. Since the lattice basis is orthogonal, vector encoding and decoding by finding appropriate nearby lattice points boils down to a simple coordinate-wise rounding procedure.

## 3  LOWER BOUNDS

We next show matching lower bounds on the communication required for MEANESTIMATION and VARIANCEREDUCTION. These results bound the number of bits a machine must receive (from any source) to output an estimate of sufficient accuracy, via an information-theoretic argument, and therefore apply to almost any model of communication. Our proofs essentially argue that, if a machine receives only a small amount of bits during the course of an algorithm, it has only a small number of possible (expected) outputs. We can therefore find instances such that our desired output ($\boldsymbol{\mu}$ or $\boldsymbol{\nabla}$) is far from any of these possible outputs. This argument is complicated, however, by the probabilistic nature of outputs (and, in the case of the VARIANCEREDUCTION problem, inputs).

**Theorem 6.** *For any* MEANESTIMATION *algorithm in which any machine receives at most $b$ bits in expectation,*

$$\mathbf{E}\left[\|\boldsymbol{EST} - \mu\|^2\right] = \Omega(y^2 2^{-\frac{3b}{d}}) \ .$$

To achieve an output variance of $O(\frac{y^2}{q})$, we see that machines must receive $\Omega(d \log q)$ bits in expectation, matching the upper bound of Theorem 2. Similarly, we have the following tight bounds for VARIANCEREDUCTION:

**Theorem 7.** *For any* VARIANCEREDUCTION *algorithm in which all machines receive (strictly) at most $b$ bits,*

$$\mathbf{E}\left[\|\boldsymbol{EST} - \boldsymbol{\nabla}\|^2\right] = \Omega(\sigma^2 n 2^{-\frac{2b}{d}}) \ .$$

This bound matches Theorem 3, since to reduce the variance expression to $O(\frac{\sigma^2}{n})$ (and, in fact, even to $O(\sigma^2)$, i.e., to achieve *any* reduction of output variance compared to input variance), we require $b = \Omega(d \log n)$ bits.

**Theorem 8.** *For any* VARIANCEREDUCTION *algorithm in which any machine receives at most $b$ bits in expectation,*

$$\mathbf{E}\left[\|\boldsymbol{EST} - \boldsymbol{\nabla}\|^2\right] = \Omega(\sigma^2 2^{-\frac{3b}{d}}) \ .$$

Here we match the leading terms of Theorem 4: to reduce variance to $O(\frac{\sigma^2}{q})$, we require $\Omega(d \log q)$ bits in expectation. Note that it is well known (and implied by e.g. Braverman et al. (2016)) that output variance cannot be reduced below $O(\frac{\sigma^2}{n})$ even with unlimited communication, so Theorem 8 implies that the full variance expression in Theorem 4 is tight.

# 4 EXPERIMENTAL VALIDATION

We investigate three distinct applications for our scheme: compressing gradients and models in variants of data-parallel SGD, and distributed power iteration for eigenvalue computation (Suresh et al., 2017). We implement the practical version of our algorithm (using the cubic lattice), which for simplicity we call RLQSGD (or LQSGD for the version without Hadamard rotation). The implemented variant ensures optimal variance, up to a $\log d$ factor (Theorem 5). We compare against QSGD (Alistarh et al., 2017), the Hadamard-based scheme of Suresh et al. (2017), as well as uncompressed baselines.

**Example 1: Compressing Gradients.** We first apply compression to a parallel solver for least-squares regression, which is given as input a matrix $A$, partitioned among the nodes, and a target vector $\boldsymbol{b}$, with the goal of finding $\boldsymbol{w}^* = \operatorname{argmin}_{\boldsymbol{w}} ||A\boldsymbol{w} - \boldsymbol{b}||_2^2$. In this example, we generate $\boldsymbol{w}^* \in \mathbb{R}^d$ and entries of $A \in \mathbb{R}^{S \times d}$ by sampling from $\mathcal{N}(0, 1)$, and we set $\boldsymbol{b} = A\boldsymbol{w}^*$. Note that the input data (and gradients) are thus naturally normalized. We then run distributed gradient descent with our quantization scheme, using 3 bits per coordinate for all methods, and examine the characteristics of the gradients sent, as well as the convergence and variance of the overall process. Figure 1 (left) shows the gradient norms and distances for the case of two nodes executing gradient descent for $d = 100$ and 8K samples, while the center and right panels show the variance of the gradient estimate for each method and convergence, respectively.

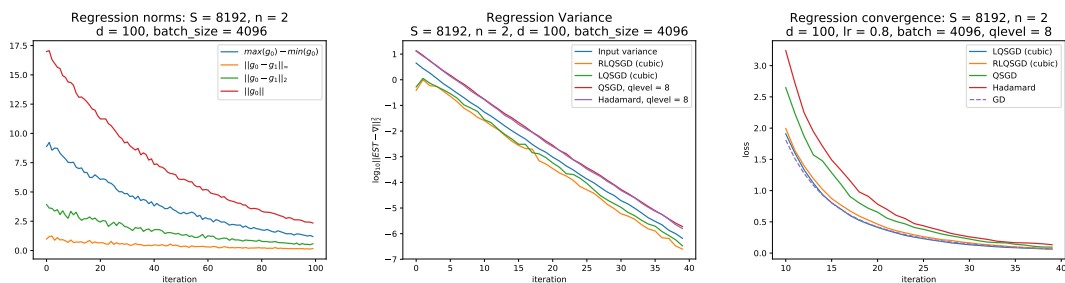

**Figure 1:** Gradient quantization results for the regression example.

Figure 1 (left) shows that, even in this simple normalized example, the distance between the two gradients throughout training is much smaller than the norm of the gradients themselves. Figure 1 (center) shows that both variants of our scheme (RLQSGD and LQSGD, with and without rotation) have significantly lower variance relative to Hadamard and standard QSGD, and in fact are the only schemes to get below input variance in this setting. Figure 1 (right) shows that this leads to better convergence using our method. The experimental report in the full version of this paper contains experiments on other datasets, and node counts, as well as a larger-scale application of our scheme to train neural networks. This application shows that our scheme matches or slightly improves the performance of specialized gradient compression schemes, at the same bit budget.

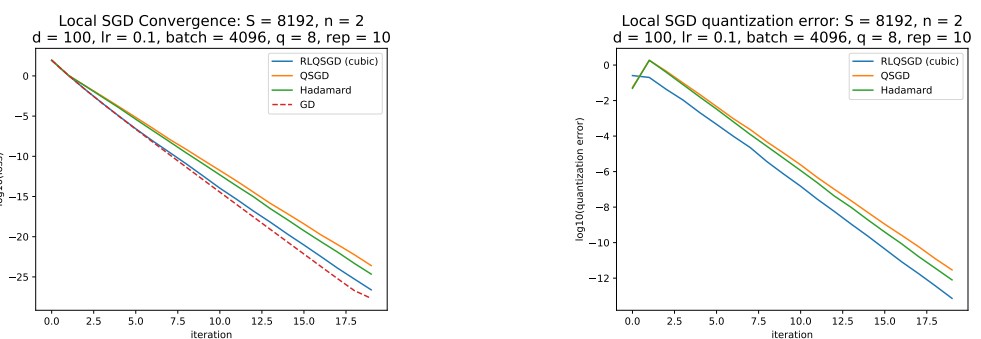

**Figure 2:** Local SGD: convergence for different quantizers (left) and quantization error (right).

**Example 2: Local SGD.** A related example is that of compressing models in LocalSGD (Stich, 2018), where each node takes several SGD steps on its local model, followed by a global *model averaging* step, among all nodes. (Similar algorithms are popular in Federated Learning (Kairouz et al., 2019).) We use RLQSGD to quantize the models transmitted by each node as part of the averaging: to avoid artificially improving our performance, we compress the *model difference* $\Delta_i$ between averaging steps, at each node $i$. RQSGD is a good fit since neither the models nor the $\Delta_i$ are zero-centered. We consider the same setup as for the previous example, averaging every 10 local SGD iterations. We illustrate the convergence behavior and the quantization error in Figure 2, which shows better convergence and higher accuracy for lattice-based quantization.

**Example 3: Distributed Power Iteration.** The power iteration method estimates the principal eigenvector of an input matrix $X$, whose rows are partitioned across machines, to form input matrices $X_i$ at the nodes. Each row of the input matrix $X$ is generated from a multivariate gaussian with first two eigenvalues large and comparable. In each iteration, each machine updates their estimate relative to its input matrix as $\boldsymbol{u}_i = X_i^T X_i \boldsymbol{x}$, and nodes average these estimates. We apply 8-bit quantization to communicate these vectors $\boldsymbol{u}_i$, following Suresh et al. (2017). Notice that our method is a good candidate here since these estimates clearly need not be zero-centered, a fact that is evident in Figure 3 (left). Figure 3 (center) shows convergence under different quantization schemes, while Figure 3 (right) shows that our method provides significantly lower quantization error across iterations.

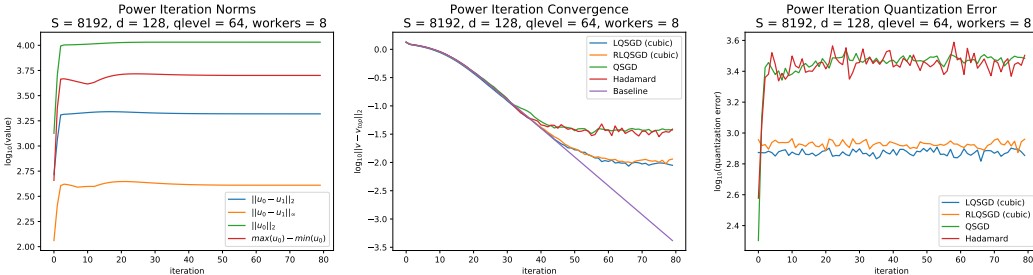

**Figure 3:** Input norms (left), convergence (center) and quantization error (right) when executing distributed power iteration on 8 parallel workers.

Overall, the results suggest that our method can leverage its support of arbitrary centering to provide consistent improvements in these three different settings.

## 5  CONCLUSIONS

We have argued in this work that for the problems of distributed mean estimation and variance reduction, one should measure the output variance in terms of the input variance, rather than the input norm as used by previous works. Through this change in perspective, we have shown optimal algorithms, and matching lower bounds, for both problems, independently of the norms of the input vectors. This improves over the theoretical guarantees provided by previous work whenever the inputs are not known to be concentrated around the origin. Our experiments suggest that this also brings about improvements in terms of practical performance. In future work, we plan to explore practical applications for variants of our schemes, for instance in the context of federated or decentralized distributed learning.

### ACKNOWLEDGMENTS

Peter Davies is supported by the European Union's Horizon 2020 research and innovation programme under the Marie Skłodowska-Curie grant agreement No. 754411. This project has also received funding from the European Research Council (ERC) under the European Union's Horizon 2020 research and innovation programme (grant agreement No. 805223 ScaleML).

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
