# OpenReview forum: "New Bounds For Distributed Mean Estimation and Variance Reduction"
_ICLR.cc/2021/Conference — ICLR 2021 Poster_

### Official Review · AnonReviewer3 · 2020-10-26
**Clean Distributed Mean Estimation approach**

**Rating:** 7
**Confidence:** 4

**Review:**

The paper considers distributed mean estimation in two variations (mean estimation and variance reduction), applicable for instance in distributed learning where several machines needs to figure out the mean of their locally computed gradients.

The paper measures the quality of an estimator in terms of the input variance, where earlier work has implicitly assumed that the input across the machines had mean zero, and instead measured quality in terms of the inputs
In that sense the approach takes in this paper generalizes previous work.

The authors provide matching upper and lower bounds for the the two problems considered, as well as a practical implementation of the general form of algorithms presented. Finally, experiments back up the quality of the approach considered.

Pros:
- I think the definition of the problems is natural and clean and the right one to consider (instead of assuming zero centered inputs).
- The approach makes application of these algorithms much simpler as the zero mean assumption is removed and does not need to be handled separately
- The general latticed based algorithms are natural and very reasonable.
- The efficient algorithm instantiation of the general approach is nice.
- It is great that the authors provide matching upper and lower bounds and in general the works seems very thorough.
- The experiments show the applicability of the general approach.

Cons:
- The actual algorithm used does not match the optimal bounds given.
- Given the nature of the problem the constants may be relevant instead of using O notation in particular in the actual algorithm presented and used in experiments.

The cons i have listed i think are all small and overall i think this is a good paper as it provides a clean practically applicable version of the problem, the bounds shown are tight and an actual new algorithm is provided and shown to have good practical qualities.

Question.
Definition 9, the packing radius. Maybe i misunderstand. Is it supposed to be the smallest r such that  two balls of radius r centered around any two different lattices points do not intersect? Because that is not what i read from the definition, but that is used in the proofs.

---

> ### Author Response · Authors · 2020-11-12
> **Individual response**
>
> **The actual algorithm used does not match the optimal bounds given.
> Given the nature of the problem the constants may be relevant instead of using O notation in particular in the actual algorithm presented and used in experiments.**
>
> Our responses to these two valid criticisms are related: as the reviewer notes, there is a difference between the “theory” and “practical” versions of our algorithms. The former achieve optimal theoretical bounds, but involve parts which in some cases can be computationally prohibitive; the latter are efficiently implementable but do not achieve the optimal theoretical bounds on worst-case inputs. For this reason, we evaluate the theoretical algorithms using asymptotic notation, but do not attempt to optimize the constants since these will not be the versions implemented in practice. Likewise, we believe the practical algorithms are best evaluated experimentally, and do not give a tight theoretical analysis of them.
>
> **Question. Definition 9, the packing radius. Maybe i misunderstand. Is it supposed to be the smallest r such that two balls of radius r centered around any two different lattices points do not intersect? Because that is not what i read from the definition, but that is used in the proofs.**
>
> The reviewer is correct - there is an error in the formal definition, which we will correct in the revision. The packing radius should be the largest (supremum) r such that two balls of radius r centered around any two different lattice points do not intersect.

---

### Official Review · AnonReviewer4 · 2020-10-27
**A result on the distributed mean estimation problem among N machines parametrized by input variance instead of input norm.**

**Rating:** 7
**Confidence:** 4

**Review:**

The paper studies the distributed mean estimation problem where $N$ machines (each holding 1 value $x_u$) wish to compute the mean of all $N$ values.
Along with values $x_u$, each machine receives a common value $y$. This value $y$ upper-bounds $\|x_u - x_v\|$ over all machines $v$. The parameter $y^2$ is called the input variance.
The authors propose a lattice-based algorithm whose quantization parameter allows a trade-off between numbers of bits needed to be communicated and the output variance of the estimated mean. One crucial contribution of the paper is that it provides guarantees with respect to input variance instead of input norm (which can be large if the inputs do not have 0 mean).


The paper is well-written. It has a clear description of the problem and provides a natural motivation. It also gives a great overview of prior work. The main idea of the algorithm is also explained in a clear way.

This work studies a basic and important problem in distributed computation. It combines and proposes several interesting ideas. I especially like the idea of lattice-based quantization and using local information, i.e., $x_v$, together with y to perform decoding.

The paper says:
-- "By choosing the leader randomly we can obtain tight bounds in expectation on the number of communication bits used per machine, and by using a more balanced communication structure such as a binary tree we can extend these bounds in expectation to hold with certainty."
Although not very crucial for this work, one should note here that using a binary tree structure would increase the number of rounds (or the communication time) by a $\log n$ factor until the final result is obtained.

Algorithms 3 and 4 perform communication setup, i.e., electing a leader or making a binary-tree like communication network. What is the communication cost of these? Is it accounted for anywhere?


Page 17, Lemma 20:
Lemma 20 implies that there EXISTS a good coloring. But why is it easy to obtain one for us to use in Algorithm 5?

--- Experiments ---
The fonts in plots are too small.
The experiments are performed only for $n=2$. It seems to be a very limiting setup. It would be nice to show experiments for larger $n$. Given that this is mainly a theoretical work, having such experiments is not a major downside, but they are also not very expressive.
Among Figures 1, 2 and 3, in only one plot the x-axis starts from 10, but in the rest from 0. Would be nice to be consistent.



--- Other comments ---

Page 1, Line 3: Should $\mu = \sum_v x_v$ be $\mu = 1/n \cdot \sum_v x_v$?

Page 2: "thanto the origin" -> "than to the origin"

Page 4, Line 5 of 2nd paragraph of Section 2.1: Is there a typo in "By choosing an appropriate of lattices"?

Page 5, first line: Would be nice to emphasize that this is a probabilistic process as $z$ is actually sampled from the linear representation of $x_u$.

Page 5, last line of statement of Theorem 1:
Should $\| z - x_u \| = O(\varepsilon)$ be $\| z - x_v \| = O(\varepsilon)$?

Page 12, 3rd line of 2nd paragraph of section A.1: Should $c_i(\lambda) = \alpha_i mod q$ be $(c_q(\lambda))_i = \alpha_i mod q$?

Page 12, 4th and 5th lines of 2nd paragraph of section A.1: $c(\lambda)$ should be $c_q(\lambda)$?

Appendix F:
There is a sentence saying:
-- "In this section we show an extension to the quantization scheme (and thereby also Algorithm 3) allowing us to use a sublinear (in d)"
But in Theorem 27, the bound is O(d log(1+q)). I do not understand why this is sublinear in d. Why does it make sense to make q=o(1)? Shouldn't q be at least a constant?

---

> ### Author Response · Authors · 2020-11-12
> **Individual response**
>
> **The paper says: -- "By choosing the leader randomly we can obtain tight bounds in expectation on the number of communication bits used per machine, and by using a more balanced communication structure such as a binary tree we can extend these bounds in expectation to hold with certainty." Although not very crucial for this work, one should note here that using a binary tree structure would increase the number of rounds (or the communication time) by a log⁡n factor until the final result is obtained.**
>
> Yes - here we chose only to consider the number of bits as our communication measure for simplicity, but in many network models one might also be concerned with round complexity, and there is a trade-off between the two measures. In the revision we will make note of this, and comment that when synchronization rounds are expensive, the one-round star-topology algorithm may be preferable.
>
> **Algorithms 3 and 4 perform communication setup, i.e., electing a leader or making a binary-tree like communication network. What is the communication cost of these? Is it accounted for anywhere?**
>
> The reviewer makes an important point - we chose to omit the costs of leader election and overlay network construction from our stated complexities, for several reasons:
> * In the settings of most interest (in particular, whenever $d > polylog(n)$), these costs will be negligible compared to those incurred by the mean estimation algorithms;
> * The exact costs will depend on the specific capabilities of the communication model, which to a large extent we can otherwise abstract;
> * There is often a trade-off between communication cost in bits, and other measures which we do not track in this work (such as round complexity);
> * These setup costs need only be incurred once, even if mean estimation is performed multiple times (as, for example, during distributed SGD);
> * Leader election and overlay construction are basic network primitives which will most likely have already been performed before our algorithms are run.
>
> As an example, in models which allow all-to-all communication, such as CONGESTED CLIQUE or MPC, we can perform leader election using only $O(1)$ expected communication bits per machine. An algorithm for doing so is the following: machines choose random IDs in $[n^3]$, and send a single bit to all other machines in the round equal to their ID, if they have not yet received a bit from any other machine. In this way, all machines are aware of the machine with the lowest ID, in $O(1) $ expected communication bits (but with an $O(n^3)$ round complexity, which would generally be considered prohibitively high). In general, in most reasonable communication models, we could perform leader election and overlay construction in, at worst, $polylog(n)$ rounds and expected bits per machine, which is dominated by the mean estimation cost when $d > polylog(n)$.
>
> In the revision we will state more explicitly that these costs are separate, and must be accounted for in whatever specific distributed system one wishes to use.

---

> > ### Author Response · Authors · 2020-11-12
> > **Individual response**
> >
> > **Page 17, Lemma 20: Lemma 20 implies that there EXISTS a good coloring. But why is it easy to obtain one for us to use in Algorithm 5?**
> >
> > The reviewer is correct to note this issue; Algorithm 5 is only the proof-of-concept “theory” version of the algorithm with error detection, and finding such a coloring to use would generally be computationally infeasible. In the full version of the paper we will detail the “practical” version of the algorithm, which uses the simpler, tractable coloring of section A.1 and is straightforwardly implementable. This algorithm, however, has weaker theoretical guarantees: it succeeds on random inputs, and on real data in all our preliminary experiments, with exponentially small failure probability, but would fail against inputs chosen adversarially against the lattice. It remains an important open question to design a practical algorithm which attains the optimal theoretical bounds against adversarial inputs.
> >
> > **The experiments are performed only for $n=2$. It seems to be a very limiting setup. It would be nice to show experiments for larger n. Given that this is mainly a theoretical work, having such experiments is not a major downside, but they are also not very expressive.**
> >
> > While most of our experiments are indeed only on two machines and focus on the effect of quantization on only a pairwise interaction, we do present experiments on 8 and 16 machines in the Appendix (Figures 12 and 13) for the real regression dataset (CPUSmall). We will aim to add some multi-machine experiments to the main body in the revision, space permitting.
> >
> >
> > **Page 5, last line of statement of Theorem 1: Should $|z−x_u|=O(\epsilon)$ be $|z−x_v|=O(\epsilon)$?**
> >
> > No; the setting of Theorem 1 is that $u$ wishes to provide $v$ with a good estimate $z$ of $x_u$, so the error of that estimate is what we wish to bound by $O(\epsilon)$.
> >
> > **Appendix F: There is a sentence saying: -- "In this section we show an extension to the quantization scheme (and thereby also Algorithm 3) allowing us to use a sublinear (in $d$)" But in Theorem 27, the bound is $O(d \log(1+q))$. I do not understand why this is sublinear in $d$. Why does it make sense to make $q=o(1)$? Shouldn't $q$ be at least a constant?**
> >
> > This is perhaps an unnatural way of expressing the communication complexities; we use it to combine the upper bounds for both sublinear and superlinear communication complexities into a single expression, and for the sublinear case we indeed take $q=o(1)$. The sublinear algorithm does not need $q$ to be at least a constant, though the linear/superlinear algorithm does. A more natural way to think about the sublinear upper bound is that $O(y^2 c^2)$ output variance uses $O(d/c)$ bits, for $c>1$, and we will clarify this in the revision.

---

### Official Review · AnonReviewer1 · 2020-10-28
**Nice to know result, which solves a particular (one may say pathological) case of an important problem, no groundbreaking techniques**

**Rating:** 6
**Confidence:** 3

**Review:**

The paper considers a particular setting of distributed mean estimation problem, where each party has a vector of potentially large $l_2$ norm, yet this vectors are fairly close to each other. The goal is to communicate as few bits as possible and estimate the mean of the vectors. Previous approaches had the dependence on the size of the ball containing all the vectors, which gives bad bounds if vectors are long (but close to each other).

The idea is to decompose each vectors into a convex hull of points in some lattice, then probabilistically round to one of these points, hash lattice points down to a short string and communicate this string. This allows to recover the points we rounded to for each party and thus estimate the mean.

In order for the communication to be efficient, the cover radius and packing radius should be within $O(1)$ from each other. For $\ell_2$ norm, this is achievable for random lattices, however such lattices are computationally intractable. The authors notice that we can reduce the original mean estimation problem to the $\ell_\infty$ case (incurring the logarithmic loss in the accuracy) and then simply use the cubic lattice.

Overall, I think the result is fairly interesting. None of the techniques are amazingly new (lattice quantization was used before in various context, e.g., locality-sensitive hashing, quantization + hashing down to a short string is a fairly standard idea as well, reduction from $\ell_2$ to $\ell_\infty$ for mean estimation was also used before as well, for, e.g., statistical queries), but I like the clean end result.

One particular request I have is to describe the computationally simple algorithm via the reduction to $\ell_\infty$ directly, not using lattices, since it's much simpler this way (it would be random rotation followed by rounding and hashing).

---

> ### Author Response · Authors · 2020-11-12
> **Individual response**
>
> **One particular request I have is to describe the computationally simple algorithm via the reduction to $\ell_{\infty}$ directly, not using lattices, since it's much simpler this way (it would be random rotation followed by rounding and hashing).**
>
> We thank the reviewer for this suggestion; we will add such a description in the revision.

---

### Official Review · AnonReviewer2 · 2020-10-30
**Reviewer2**

**Rating:** 6
**Confidence:** 4

**Review:**

Summary: This paper studies the problem of mean estimation of n vectors in R^d in a distributed setting. There are n machines. Each has one data point, a vector x_v. The goal is to compute the mean of x_v’s for v = 1, …, n with as few bits of communication as possible.
Their main contribution is using a new quantization method by exploiting the structure of Lattices. The idea is that each machine randomly assigns its point to a nearby lattice point. Then, the lattice point can be expressed by a few bits. While two points might be assigned to the same bit strings (since the lattice has infinitely many points), the hope is that these points are far apart from each other.
Furthermore, we need to ensure that other machines could decode the bit string and obtain the same lattice point again and use this point to compute the average. Thus, another machine would interpret the bit string as a close-by lattice point. Note that since we assume some guarantees that all the x_v's are close to each, it suffices that a machine selects the closest lattice point that its description would match the bit string.
Experimental evaluation is also provided.

Overall evaluation: The authors consider an essential problem and use an interesting idea to solve it. Adding more discussion and literature review might help improve the paper. The writeup could be improved as well.

Major comments:
- It would be beneficial if the authors discuss and compare the following straightforward approach: We can use a previously known algorithm and allow the machines to obtain a potentially inaccurate estimate of the mean, say \hat \mu. Then, we can run the algorithm again. This time vectors being x_v - \hat \mu. In this step, we bring down the points closer to the center. Thus, the accuracy increases over time. We can repeat this process several times until we achieve the desired accuracy.

- It might be worth looking at a series of papers that studied the mean estimation of random vectors in a non-distributed setting. They mainly focus on the tail behavior of the estimate. They use different estimates other than average, such as the median of means. See “On the estimation of the mean of a random vector” and more recent papers that cited this one.

Minor comments:
- It is unclear from the main text which convex hull is picked. Maybe it worth discussing some high-level explanations in the main text as well.
- Bottom of page 4: Using Chebyshev’s inequality would only guarantee that a constant fraction of points is within distance O(\sigma \sqrt{n}).
- Abstract, Line 3: where \mu is defined, “\n” is missing.
- Page 2, last paragraph: “thanto” -> other than the
- Page 7, Line 1: O(\sigma^2) -> Shouldn’t be O(\sigma^2/n)?

---

> ### Author Response · Authors · 2020-11-12
> **Individual response**
>
> **It would be beneficial if the authors discuss and compare the following straightforward approach: We can use a previously known algorithm and allow the machines to obtain a potentially inaccurate estimate of the mean, say $\hat \mu$. Then, we can run the algorithm again. This time vectors being $x_v - \hat \mu$. In this step, we bring down the points closer to the center. Thus, the accuracy increases over time. We can repeat this process several times until we achieve the desired accuracy.**
>
> This is an interesting proposal. We first note that, if we applied a previously known algorithm, we would still obtain error bounds in terms of input norms rather than differences, due to the first iteration, so we would not see a direct comparison. The comparative performance would, as currently, depend on the disparity between input norms and differences.
>
> However, we could also apply our own algorithm in this fashion to compare the performance: in this case, we see that we would obtain the same asymptotic communication bounds by iterating as we would by reaching the target accuracy in one iteration. To see this, consider two iterations, in which we first reduce input variance by a multiplicative factor of $q_1$, and then by a factor of $q_2$, overall decreasing variance by $q = q_1 q_2$. In total we use $O(d \log q_1 + d \log q_2) = O(d \log q)$ bits per machine, matching the complexity of our current one-shot algorithm.
>
> **It might be worth looking at a series of papers that studied the mean estimation of random vectors in a non-distributed setting. They mainly focus on the tail behavior of the estimate. They use different estimates other than average, such as the median of means. See “On the estimation of the mean of a random vector” and more recent papers that cited this one.**
>
> Since our theoretical work is so far only concerned with asymptotic results, we were content to approximate sample mean (which is an asymptotically optimal estimator for any reasonable class of distributions) and concentrate on minimizing the additional error incurred by quantization. However, the reviewer is quite right - if one aims to minimize the constant factor in the error, there is a rich line of research from statistics into better estimators of the mean. Using these in a distributed fashion would add another layer of technical difficulty, since they are not necessarily aggregate functions and so could not be combined over a binary tree as in Algorithm 4. We will add discussion of this in the revision.
>
> **It is unclear from the main text which convex hull is picked. Maybe it worth discussing some high-level explanations in the main text as well.**
>
> Any convex hull of lattice points containing the input vector suffices for the theoretical results; procedures for finding such convex hulls will depend on the lattice and norm chosen. We will add discussion of what this means for the cubic lattice in particular (where such a convex hull can be found by a simple rounding procedure).
>
> **Bottom of page 4: Using Chebyshev’s inequality would only guarantee that a constant fraction of points is within distance $O(\sigma \sqrt{n})$.**
>
> This statement in the paper is aimed primarily at providing an informal intuition, but what we mean by it is the following:
> Chebyshev’s inequality implies that an input is within distance $c \sigma \sqrt{n}$ of the true vector ${\mathcal \nabla}$ with probability at least $1 - 1/(c^2 n)$. So, by a union bound we have that with probability $1-1/(c^2)$ all inputs are within distance $c \sigma \sqrt{n}$ of ${\mathcal \nabla}$, and (setting $c$ to be a sufficiently large constant) therefore within $O(\sigma \sqrt{n})$ distance of each other.
>
> **Page 7, Line 1: $O(\sigma^2)$ -> Shouldn’t be $O(\sigma^2/n)$?**
>
> No, although we should perhaps rephrase for clarity in the revision. The message of that line is that Theorem 7 implies (possibly surprisingly) that in order to agree even on an output with $\Theta(\sigma^2)$ output variance, we need $\Omega(d \log n)$ bits, the same asymptotic amount required to reach optimal $O(\sigma^2/n)$ output variance.

---

### Author Response · Authors · 2020-11-12
**General revision plan**

We thank the reviewers for their time and helpful comments. We aim to provide individual responses addressing the points of each review, highlighting the changes we will make in the revision to reflect these points. Minor comments requiring straightforward corrections are omitted from these responses, but will naturally correct in the revision. We thank the reviewers for drawing attention to them.

We plan to incorporate all the mentioned changes in the revision, and would welcome and greatly appreciate further comment from reviewers if they feel there are issues which we still have not adequately addressed, so that we may make further improvements.

---

### Decision · Program_Chairs · 2021-01-07
**Final Decision**

**Decision:**

Accept (Poster)

**Comment:**

This paper presents a new algorithm for distributed multivariate mean estimation. This method performs significantly better than previous approaches when the input vectors have large norm but are relatively close to each other. The approach relies on lattices and randomized rounding. The approach is evaluated experimentally as well. Overall, there is consensus among the reviewers that this work solves a clean problem using non-trivial ideas. I recommend accepting the paper.